# GInX-Eval: Towards In-Distribution Evaluation of Graph Neural Network Explanations

## Abstract

Diverse explainability methods of graph neural networks (GNN) have recently been developed to highlight the edges and nodes in the graph that contribute the most to the model predictions. However, it is not clear yet how to evaluate the *correctness* of those explanations, whether it is from a human or a model perspective. One unaddressed bottleneck in the current evaluation procedure is the problem of out-of-distribution explanations, whose distribution differs from those of the training data. This important issue affects existing evaluation metrics such as the popular faithfulness or fidelity score. In this paper, we show the limitations of faithfulness metrics. We propose **GInX-Eval** (**G**raph **In**-distribution e**X**planation **Eval**uation), an evaluation procedure of graph explanations that overcomes the pitfalls of faithfulness and offers new insights on explainability methods. Using a fine-tuning strategy, the GInX score measures how informative removed edges are for the model and the HomophilicRank score evaluates if explanatory edges are correctly ordered by their importance and the explainer accounts for redundant information. GInX-Eval verifies if ground-truth explanations are instructive to the GNN model. In addition, it shows that many popular methods, including gradient-based methods, produce explanations that are not better than a random designation of edges as important subgraphs, challenging the findings of current works in the area. Results with GInX-Eval are consistent across multiple datasets and align with human evaluation.

## 1 Introduction

While people in the field of explainable AI have long argued about the nature of good explanations, the community has not yet agreed on a robust collection of metrics to measure explanation *correctness*. Phenomenon-focused explanations should match the ground-truth defined by humans and are evaluated by the accuracy metric. Model-focused explanations contribute the most to the model predictions and are evaluated by the faithfulness metrics. Because ground-truth explanations are often unknown, faithfulness and its variants are the most common measures of quality. Faithfulness metrics remove or retrain only the important graph entities identified and observe the changes in model outputs. However, this edge masking strategy creates Out-Of-Distribution (OOD) graph inputs, so it is unclear if a high faithfulness score comes from the fact that the edge is important or from the distribution shift induced by the edge removal (Günnemann, 2022).

We propose **GInX-Eval**, an evaluation procedure of in-distribution explanations that brings new perspectives on GNN explainability methods. Testing two edge removal strategies, we evaluate the impact of removing a fraction $t$ of edges in the graph on the GNN model performance. To overcome the OOD problem of faithfulness metrics, the model is fine-tuned and tested on the reduced graphs at each degradation level. The best explainability methods can identify the graph entities whose removal triggers the sharpest model accuracy degradation. We compare generative and non-generative methods on their **GInX** score against a random baseline across four real-world graph datasets and two synthetic datasets, all used for graph classification tasks. With this strategy, we show that existing explainers are not better than random in most of the cases. In addition, we show the overall superiority of GNNExplainer, PGMExplainer, and most of the generative methods above gradient-based methods and Occlusion. Our results lead to diverging conclusions from recent studies (Yuan

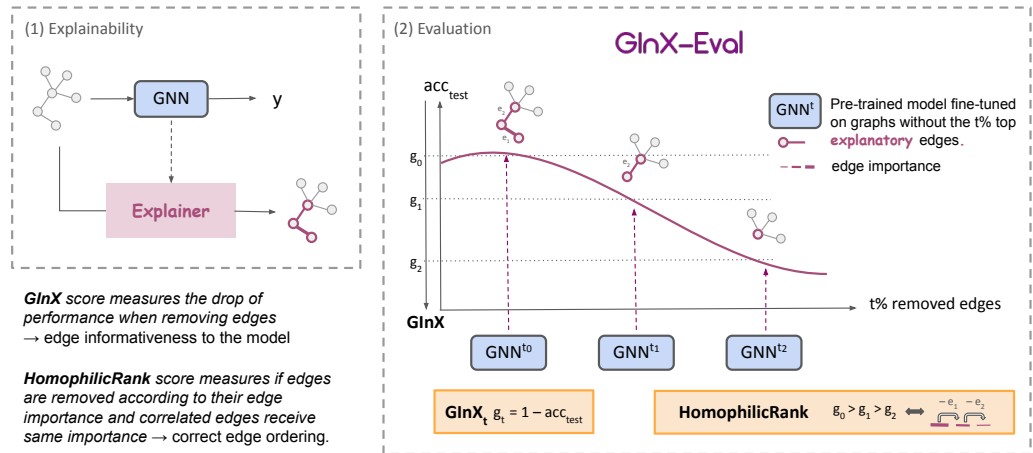

Figure 1: Summary of GInX-Eval procedure. (1) A GNN model is pre-trained to predict the class of the input graphs. An explainability method generates explanatory subgraphs. (2) For each $t \in [0.1, ..., 0.9]$, a new train and test datasets are generated where the fraction $t$ of the top explanatory edges is removed. At each $t$, the pre-trained GNN model is fine-tuned on the new train dataset, evaluated on the new test set, and the GInX score is computed. If the model performance decreases, i.e., the GInX scores increase, the explanatory edges are informative to the model. The HomophilicRank score is also computed to evaluate if explanatory edges are correctly ranked by the explainability method and redundant explanatory edges are accounted for.

et al., 2023; Agarwal et al., 2022) and again question the use of faithfulness as a standard evaluation metric in explainable AI (xAI). The GInX-Eval framework also proposes the **HomophilicRank** score to assess the capacity of explainers to correctly order edges by their true importance for the model and value the same correlated important edges. Finally, GInX-Eval is a useful tool to validate ground-truth explanations provided with some datasets and discover both human- and model-based explanations. Because it relies on a fine-tuning strategy of black-box pre-trained models, GInX-Eval is also a useful evaluation procedure in real-world scenarios where models are not publicly available and can only be used via API calls. Due to the computational cost of re-training, GInX-Eval is not proposed as a systematic evaluation metric but as a tool to throw light on the true informative power of explainers and validate ground-truth explanations. To summarize our contributions:

- We first show that faithfulness evaluates OOD explanations. In addition, we observe that (1) it is inconsistent with the accuracy metric, (2) it leads to divergent conclusions across datasets, and (3) across edge removal strategies. Overcoming the OOD problem, we propose **GInX-Eval** (**G**raph **In**-distribution e**X**planation **Eval**uation), a new evaluation framework for GNN explainability methods. The **GInX** score evaluates how informative explanatory edges are to the model and the **HomophilicRank** score assesses if those edges are correctly ordered by their importance and correlated edges share a similar importance weight.

- We propose a validation protocol of ground-truth explanations using the GInX score. This way, we can measure the degree of alignment between human-based and model-based explanations.

- With GInX-Eval, we now finally demonstrate the true informative power of well-known explainability methods, filter out bad methods, and choose methods that can correctly rank edges.

The rest of this article is organized as follows. Section 2 discusses the literature on graph neural networks (GNN) explainability evaluation and the OOD problem. Section 3 presents the limitations of the current evaluation with faithfulness and introduces GInX-Eval, our new in-distribution evaluation procedure, and its two scores, GInX and HomophilicRank. Section 4 presents the experiments that we conducted in detail. Section 5 summarizes the paper and discusses the future opportunities.

## 2 RELATED WORK

**Evaluation in xAI.** To measure the correctness of explanations, a few metrics have been developed. GNN explainability method should satisfy accuracy, faithfulness, stability (Sanchez-Lengeling et al.; Yuan et al., 2023; Agarwal et al., 2021; 2022), consistency and contrastivity (Yuan et al., 2023), usefulness (Colin et al., 2021). The two most popular approaches are: (1) measuring accuracy using ground-truth annotations and (2) measuring faithfulness using objective metrics (Chan et al., 2022). However, the accuracy metric also referred to as plausibility (Li et al., 2022; Longa et al., 2022; Nauta et al., 2022), needs ground-truth explanations and is therefore less universal and more subjective than faithfulness. Nauta et al. (2022) argues that evaluating the accuracy of an explanation to humans is different from evaluating its correctness. It is not guaranteed that it aligns with its faithfulness Jacovi & Goldberg (2020). According to Petsiuk et al. (2018), it is preferable to keep humans out of the evaluation to better capture the model's understanding rather than representing the human's view. Faithfulness metrics are therefore the most popular evaluation metrics, but we show later in Section 3.2 that they have strong limitations including evaluating out-of-distribution explanations.

**Solving the OOD Problem.** Recent works have proposed to adapt the GNN model or develop robust explainability methods to overcome the out-of-distribution (OOD) problem. Faber et al. (2020) argue that explanations should stay in the training data distribution and propose CoGe to produce Distribution Compliant Explanation (DCE). Li et al. (2021) propose a novel out-of-distribution generalized graph neural network. Hsieh et al. (2020) do not remove features but apply small adversarial changes to the feature values. Instead of developing robust methods, Hooker et al. (2019) evaluate interpretability methods by observing how the performance of a retrained model degrades when removing the features estimated as important. While this retraining strategy circumvents the OOD problem, it has only been developed for CNN models on images to evaluate feature importance. Building on this retraining strategy, we propose a new evaluation procedure for GNN models and introduce an alternative to faithfulness metrics.

## 3 METHOD

This section highlights the limitations of the popular xAI evaluation procedure using faithfulness metrics and proposes GInX-Eval to overcome those. We can assess the informativeness of explanations for the model with the GInX score and the capacity of methods to correctly order explanatory edges by their importance and identify correlated edges with the HomophilicRank score.

### 3.1 PRELIMINARIES

Given a well-trained GNN model $f$ and an instance of the dataset, the objective of the explanation task is to identify concise graph substructures that contribute the most to the model's predictions. The given graph can be represented as a quadruplet $G(\mathcal{V}, \mathcal{E}, \mathbf{X}, \mathbf{E})$, where $\mathcal{V}$ is the node set, $\mathcal{E} \subseteq \mathcal{V} \times \mathcal{V}$ is the edge set. $\mathbf{X} \in \mathbb{R}^{|\mathcal{V}| \times d_n}$ and $\mathbf{E} \in \mathbb{R}^{|\mathcal{E}| \times d_e}$ denote the feature matrices for nodes and edges, respectively, where $d_n$ and $d_e$ are the dimensions of node features and edge features. In this work, we focus on structural explanation, *i.e.,* we keep the dimensions of node and edge features unchanged. Given a well-trained GNN $f$ and an instance represented as $G(\mathcal{V}, \mathcal{E}, \mathbf{X}, \mathbf{E})$, an explainability method generates an explanatory edge mask $M \in \mathbb{R}^{|\mathcal{E}|}$ that is normalized. Furthermore, to obtain a human-intelligible explanation, we transform the edge mask to a sparse matrix by forcing to keep only the fraction $t \in \mathcal{T}$ of the highest values and set the rest of the matrix values to zero. Each explainability method can be expressed as a function $h : \mathcal{G} \to \mathcal{G}$ that returns for each input graph $G$ an explanatory subgraph $h(G)$.

**Edge Removal Strategies.** There are two strategies to select a fraction $t \in \mathcal{T}$ of edges in a graph: the *hard* selection and the *soft* selection. The hard selection function $\chi_H : \mathcal{G} \times \mathcal{T} \to \mathcal{G}$ picks a fraction $t$ edges from a graph $G$ so that the number of edges and nodes is reduced. Applying the hard selector on an explanation $G' = h(G)$ for a fraction $t$ of edges, we obtain a *hard* explanation $\chi_H(G', t) = G'(\mathcal{V}', \mathcal{E}', \mathbf{X}', \mathbf{E}')$, such that $\mathcal{V}' \subseteq \mathcal{V}$ and $\mathbf{X}' = \{X_j | v_j \in \mathcal{V}'\}$, where $v_j$ and $X_j$ denote the graph node and the corresponding node features. The hard explanation has only the nodes connected to the remaining important edges and is usually smaller than the input graphs that very

likely do not lie in the initial data distribution. The soft selection function $\chi_S : \mathcal{G} \times \mathcal{T} \rightarrow \mathcal{G}$ instead sets edge weights to zero when edges are to be removed. Therefore it preserves the whole graph structure with all nodes and edge indices. Given an explanation $G'$ and the $t$-sparse explanatory mask $M_t$ that keeps the $t\%$ highest values in $M$ and sets the rest to zero, we can express the *soft* explanation at $t$ as $\chi_S(G', t) = G'(\mathcal{V}, \mathcal{E}, \mathbf{X}, \mathbf{E}, M_t)$. It has similar edge index and nodes as the input graph $G$ but unimportant edges receive zero weights. Note here that, unlike the soft removal strategy, the hard removal strategy might break the connectivity of the input graphs, resulting in explanations represented by multiple disconnected subgraphs.

## 3.2 FAITHFULNESS METRICS

**Definition**  The faithfulness or fidelity scores are the most general quality metrics in the field of GNN explainability. To evaluate the correctness of the explanation, the explanatory subgraph or weighted graph $h(G)$ produced by the explainer $h$ is given as input to the model to compute the fidelity score on the probabilities:

$$fid = |p(f(h(G)) = y) - p(f(G) = y)| \in [0; 1] \tag{1}$$

where $y$ is the true label for graph $G$ and $f(G)$, $f(h(G))$ the predicted labels by the GNN given $G$ and $h(G)$ respectively. The closer $fid$ is to 0, the more faithful the explanation is. The faithfulness score is averaged over the N explanatory graphs $h(G_i), i \leq N$ as:

$$\text{Faithfulness} = 1 - \frac{1}{N} \sum_{i=1}^{N} |p(f(h(G_i)) = y_i) - p(f(G_i) = y_i)| \in [0; 1] \tag{2}$$

The metric is normalized and the closer it is to 1, the more faithful the evaluated $N$ explanations are to the initial predictions. The above score corresponds to the $fid^{-prob}$, one of the four forms of the fidelity scores Yuan et al. (2023), described in Appendix A.1.

**Prior Work.**  While faithfulness metrics are the most popular quality measures independent of ground-truth explanations, they have been recently criticized. Based on a "removal" strategy, i.e., we remove or keep the graph entities estimated as important, faithfulness withdraws some entities by setting them to a baseline value either removing them from the graph or putting their weight to zero. Hsieh et al. (2020) correctly observe that this evaluation procedure favors graph entities that are far away from the baseline. Consequently, methods that focus on highly weighted edges while maybe ignoring low-weight but still essential connections are favored. In addition, truncated graphs after edge removal can lie out of the data distribution used for training the GNN model (Hooker et al., 2019). In this case, model behavior might differ not because of removing important information but because of evaluating a sample outside the training distribution. The out-of-distribution risk is even larger with graphs because of their discrete nature (Faber et al., 2020).

## 3.3 GINX-EVAL

GinX-Eval is an evaluation procedure of explainability methods that overcomes the faithfulness metrics' OOD problem and assesses the informativeness of explanatory edges towards the GNN model. Figure 1 gives an overview of the procedure. To evaluate the explainer $h$, GInX-Eval first gathers the explanations produced by $h$ for all graph instances. The explanatory edges can be ranked according to their respective weight in the subgraph: the most important edges have a weight close to 1 in the mask, while the least important ones have weights close to 0. At each degradation level $t$, we remove the top $t$ fraction of the ordered explanatory edge set from the input graphs. We generate new train and test sets at different degradation levels $t \in [0.1, 0.2, ..., 1]$. The pre-trained GNN model is then fine-tuned at each degradation level on the new train dataset and evaluated on the new test data. While being the most computationally expensive aspect of GInX-Eval, fine-tuning is scalable (see Appendix B.4) and we argue that it is a necessary step to decouple whether the model's degradation in performance is due to the loss of informative edges or due to the distribution shift. The objective here is not to provide a computationally efficient evaluation metric but to highlight the limitations of popular evaluation in xAI for GNN and question the superiority of gold standard methods. The pseudo-code to implement GInX-Eval is given in Appendix B.3.

A drop in test accuracy when removing edges indicates that those edges were important for the model to make correct predictions. These edges are therefore considered as important as they are the most informative to the model. It is worth noticing that edges might be correlated and those spurious correlations can lead to an absence of accuracy drop when removing the top important edges and then a sudden decrease in accuracy when all correlated edges are removed.

### 3.3.1 GInX Score

Following this evaluation procedure, we define the GInX score at $t$. It captures how low the test accuracy is after removing the fraction $t$ of edges. Let $h(G)$ be the explanatory subgraph generated by the method $h$, $y$ the true label for graph $G$ and $\chi : \mathcal{G} \times \mathcal{T} \to \mathcal{G}$ the edge selection function irrespective of the edge removal strategy we use. $\chi$ takes an explanation $h(G)$ and returns the hard or soft explanatory graph containing the fraction $t \in \mathcal{T}$ of the explanatory edges according to the explainer $h$. We define GInX($t$) as:

$$ \text{GInX}(t) = 1 - \text{TestAcc}(t) = 1 - \frac{1}{N_{test}} \sum_{i=0}^{N_{test}} \mathbb{1}(f(G_i \setminus \chi(h(G_i), t)) = y_i) \qquad (3) $$

The closer the GInX score is to one, the more informative the removed edges are to the model. Note here that the GInX score at $t$ can be computed following the hard or soft edge removal strategy; however, we show in Appendix A.2 that the GInX score computed with hard edge removal has higher expressiveness.

### 3.3.2 HomophilicRank Score

Based on the GInX score, we can compute the power of explainability methods to rank edges, i.e., to correctly order edges based on their importance and give identical importance to correlated edges. The edge homophilic ranking power can be evaluated with the HomophilicRank score defined as:

$$ \text{HomophilicRank} = \sum_{t=0,0.1,...,0.8} (1 - t) \times (\text{GInX}(t + 0.1) - \text{GInX}(t)) \qquad (4) $$

The HomophilicRank score measures the capacity of a method to rank edges by their correct importance ordering while assigning similar importance weights to correlated edges. A high score characterizes methods that (1) assign the highest importance weights to the most genuinely informative edges for the model and (2) treat correlated edges on equal footing. It measures the capacity to uniformly assign importance to redundant information rather than only putting importance to a single representative edge and none to the correlated edges. This score penalizes methods that for instance only discover a subset of important edges and do not account for their correlated edges. This is especially important when you try to characterize an explanation and identify fundamental entities within the explanatory substructure.

## 4 Experimental results

In the following section, we propose to validate the GInX-Eval procedure and show its superiority over the widely used faithfulness metric. We support our claims with well-chosen experiments.

**Experimental Setting** Explainability methods were evaluated on two synthetic datasets, BA-2Motifs and BA-HouseGrid, three molecular datasets MUTAG, Benzene and BBBP, and MNISTbin. They all have ground-truth explanations available except for the BBBP dataset. We test three GNN models: GCN Kipf & Welling (2017), GAT (Veličković et al., 2018), and GIN (Hu et al., 2020) because their scores are high on the selected real-world datasets, with a reasonable training time and fast convergence. For the two synthetic datasets, we only use GIN since the GCN and GAT models do not give good accuracy. Further details on the datasets, GNN training parameters, and time are given in Appendix B. We compare non-generative methods, including the heuristic Occlusion (Zeiler

& Fergus, 2014), gradient-based methods Saliency (Baldassarre & Azizpour, 2019), Integrated Gradient (Sundararajan et al., 2017), and Grad-CAM (Pope et al., 2019), and perturbation-based methods GNNExplainer (Ying et al., 2019), PGMExplainer (Vu & Thai, 2020) and SubgraphX (Yuan et al., 2021). We also consider generative methods: PGExplainer (Luo et al., 2020), GSAT (Miao et al., 2022), GraphCFE (CLEAR) (Ma et al., 2022), D4Explainer and RCExplainer (Bajaj et al., 2021). For more details on the differences between generative and non-generative explainers, we refer the reader to Appendix B.5. We compare those explainability methods to base estimators: Random, Truth, and Inverse. Random assigns random importance to edges following a uniform distribution. Truth estimates edge importance as the pre-defined ground-truth explanations of the datasets. The Inverse estimator corresponds to the worst-case scenario where edges are assigned the inverted ground-truth weights. If $w_{i,j}$ is the ground-truth importance of the edge connecting nodes $i$ and $j$, the weight assigned by the Inverse estimator is equal to $1 - w_{i,j}$.

## 4.1 THE OUT-OF-DISTRIBUTION FAITHFULNESS EVALUATION

The biggest limitation of the faithfulness metrics is the so-called OOD problem. The generated explanations are out-of-distribution, i.e. they lie outside the data distribution and "fool" the underlying predictor to change the original class, i.e., $f(h(G)) \neq f(G)$. Whereas, in factual explainability scenarios, we expect the explanatory graph $h(G)$ to have the same class as the input graph $G$, i.e., $f(h(G)) = f(G)$. Figure 2 illustrates the OOD problem: the extracted model embeddings of explanations of toxic molecules are more similar to the ones of non-toxic molecules. In this case, the model predicts the explanatory subgraphs to be non-toxic while they are valid toxic molecular fragments. The model prediction is altered not necessarily because we keep only the important entities but also because the model lacks knowledge about these new explanatory graphs. Therefore, the faithfulness score which definition is based on the model predictions of explanations, does not entirely capture the quality of explanations and is ill-suited to evaluate explainability methods.

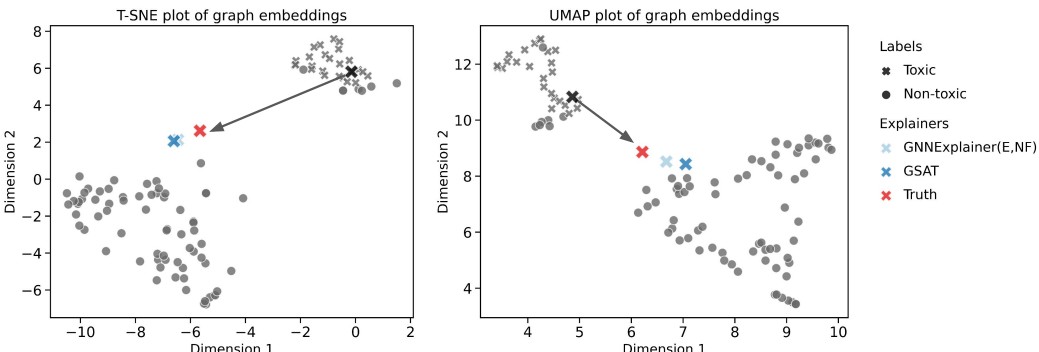

Figure 2: Illustration of the out-of-distribution problem: explanations of a toxic molecule lie closer to the non-toxic molecular representations. Graph embeddings were extracted after the readout layer of the pre-trained GIN model for the MUTAG dataset. We use both t-SNE and UMAP to project the embeddings into 2D representations. Both projection methods show the existence of out-of-distribution explanations.

As a result, we cannot rely on the evaluation with faithfulness to draw general conclusions on the explainability methods. We compare the rankings of explainability methods according to the faithfulness evaluated on the two types of explanations, Hard Fidelity and Soft Fidelity respectively, and the accuracy score defined as the AUC score to stay consistent with previous work Longa et al. (2022). The AUC score is computed between the explanatory weighted edge mask and the ground-truth edge mask with binary values in $\{0, 1\}$.

**Observation 1** *The faithfulness metric is not consistent with the accuracy score.* In figure 3, there is a general misalignment in the rankings of explainers and base estimators on faithfulness or AUC score. For all datasets but Benzene, the Truth estimator, whose accuracy is maximal, has a small faithfulness score $\sim 0.5$. For MNISTbin, Inverse is by far the best according to the faithfulness

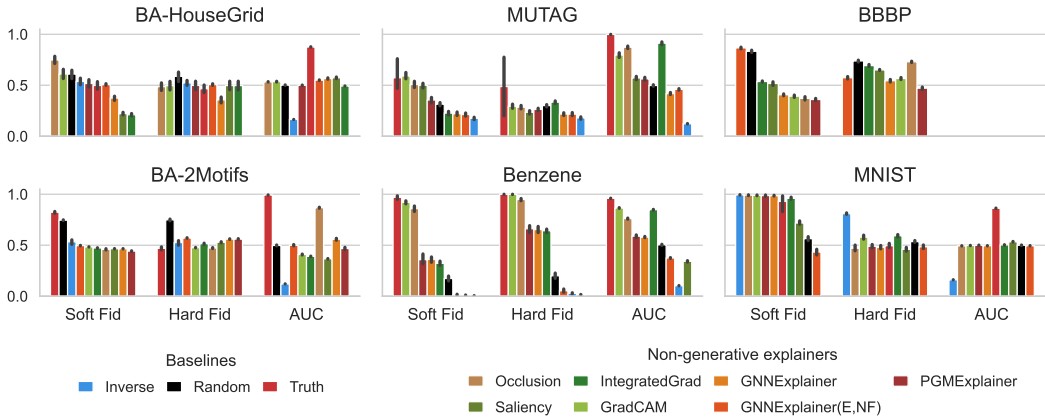

Figure 3: Rankings of base estimators and non-generative explainability methods according to the faithfulness score computed on soft explanations, the faithfulness score on hard explanations, and the AUC score. The AUC ranking is only reported for datasets with ground-truth explanations. Baselines were evaluated on the full explanatory masks, while explainability methods were evaluated on the truncated explanations, keeping the top 10 important undirected edges.

score while being the worst explainer by definition on the AUC score. For BA-2Motifs, Random has the highest faithfulness score but can only be 50% accurate by definition. Due to the OOD problem of faithfulness, we cannot decide if the model is fooled by the subgraphs induced by the most informative edges or if human-based and model-based evaluations disagree. Therefore, we cannot quantify the alignment between human and model explainability.

**Observation 2** *The evaluation of the explainability methods with the faithfulness metric is not consistent across datasets.* Figure 3 shows no consensus on the top-3 methods across datasets on the soft faithfulness or hard faithfulness score. For instance, we observe that GradCAM and Occlusion have the highest Soft Fid scores for BA-House-Grid, MUTAG, Benzene, and MNISTbin, but not for BA-2Motifs and BBBP where Truth, Random, and GNNExplainer outperform. For Hard Fid, the results are also very heterogeneous among the six datasets. Due to the OOD problem, we cannot conclude that those inconsistencies across datasets are related to differences inherent to the graph data itself, e.g., differences in graph topology, size, or density among the datasets.

**Observation 3** *The faithfulness metric is not consistent across edge removal strategies.* On figure 3, the top-3 ranking for Soft Fid and Hard Fid is always different except for Benzene dataset. This means that the edge removal strategy influences the model perception: the model does not predict labels only based on the information contained in the explanatory edges but also based on the structure of the given explanations. Because of the OOD problem, we cannot decide whether those inconsistencies come from the explainability methods themselves: methods that produce disconnected explanations are penalized by the hard removal strategy because the GNN model is not able to process the message passing.

## 4.2 VALIDATION OF GINX-EVAL PROCEDURE

We validate the GInX-Eval procedure on the BA-HouseGrid synthetic dataset because ground-truth explanations, i.e., house and grid motifs, are very well-defined and class-specific. In the binary classification setting, graphs are labeled 1 if they have grids and 0 if they have house motifs attached to the main Barabási graph. We test three explainability baselines: the Random explainer that assigns values in $[0, 1]$ following a uniform distribution, the Truth that assigns ground-truth explanations, and the Inverse estimator that returns the inverse ground-truth explanations and is, therefore, the worst estimator possible.

On Figure 4, GInX-Eval distinguishes the three methods because we observe a sharp decrease of the Truth explainer after 10% edge removal, while the Inverse estimator does not degrade the model performance, and the Random baseline starts decreasing after 20% of the edges are removed. Without re-training, all base importance estimators lead to a model performance degradation. Therefore, evaluating without retraining the model cannot reflect the true explainability power of the methods.

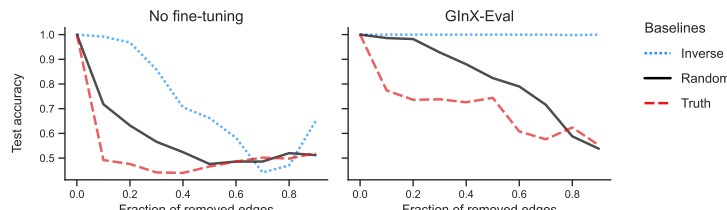

Figure 4: Comparison between not fine-tuning the GNN model and GInX-Eval on the BA-HouseGrid dataset. Without fine-tuning, the model's performance also decreases for the worst estimator Inverse where uninformative edges are removed first, preventing a correct evaluation of explainability methods. However, for GInX-Eval where the model is fine-tuned on modified datasets, we observe no test accuracy degradation for the worst-case estimator Inverse.

### 4.3 EVALUATING WITH GINX-EVAL

#### 4.3.1 OVERVIEW

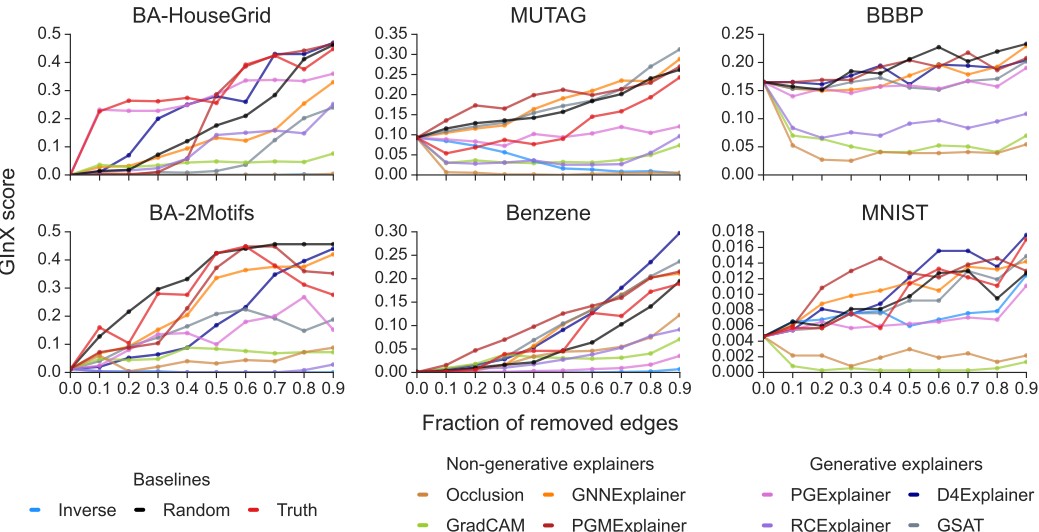

Figure 5: GInX scores of a fine-tuned GIN model on graphs with increasing fraction of removed edges. The removed edges are the most important based on explainability methods, and new input graphs are obtained with the *hard* selection, i.e., explanatory edges are strictly removed from the graph so that the number of edges and nodes is reduced. For more methods, see Appendix C.1.

GInX-Eval evaluates to what extent removing explanatory edges degrades the model accuracy. We adopt the hard selection strategy to remove edges. Even if conclusions are similar for both selection strategies (see Appendix A.3 and C.2), the degradation is of the order of $10^{-1}$ with hard selection versus $10^{-2}$ for soft selection. For visualization purposes, we prefer conveying here results with the hard selection. We refer the reader to Appendix C.2 for additional results with the soft selection.

Figure 5 shows how the GInX score increases when we remove a fraction $t \in [0.1, ..., 0.9]$ of the most important edges according to multiple explainability methods. For clarity, we choose to display a smaller set of explainability methods. For more methods, we refer the reader to Appendix C.1. We first observe that model performance is remarkably robust to graph modification for the four real-world datasets, with a maximum growth of the GInX score of 30% observed for the Benzene dataset. For the synthetic datasets, removing many edges leads to a random assignment of labels by the model. In real-world datasets, GNN models might be able to capture high-level information even with absent connections.

We note a particularly small increase of the GInX score for MNISTbin, i.e., in the order of $10^{-2}$. For this dataset, the GNN model is robust to edge modification. After removing most of the edges from the input graph, the model retains most of the predictive power. The reason might be that node

Table 1: Truth mask sparsity values for each dataset and the deduced optimal thresholds.

| Dataset | Truth sparsity | Optimal threshold |
|---|---|---|
| **BA-2Motifs** | 0.216 | 0.3 |
| **BA-HouseGrid** | 0.065 | 0.1 |
| **Benzene** | 0.175 | 0.2 |
| **MNISTbin** | 0.235 | 0.3 |
| **MUTAG** | 0.039 | 0.1 |

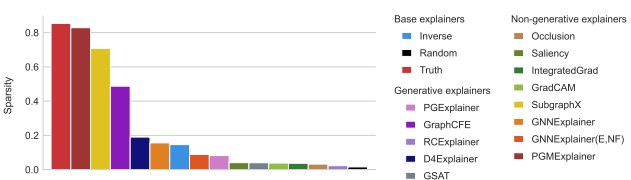

Figure 6: Mask sparsity of different explainability methods. A high sparsity indicates an explanatory mask with many zeros and small pre-processing explanatory subgraphs.

and edge features are more important for the prediction than the graph structure itself for those two datasets.

### 4.3.2 GINX-EVAL OF BASE ESTIMATORS

*Is the ground-truth explanation meaningful to the model?* The Truth and the Inverse edge importance estimators are evaluated on all datasets except BBBP which has no ground-truth available. We observe in figure 5 that the GInX score stays constant for Inverse and drops significantly for Truth. We conclude that the explanations generated with Inverse have only uninformative edges for the model, while the ground-truth edges contain crucial information for making correct predictions. GInX-Eval is a useful tool to validate the quality of provided ground-truth explanations of published graph datasets.

*Does a random assignment of edge importance informative to the model?* For all datasets except Benzene, the Random baseline leads to a similar degradation as the Truth estimator in figure 5. There are two reasons for this. First, random explanations contain a few edges present in the ground-truth explanation. Removing just these few edges makes the GInX score increase sharply because of the strong correlations that exist among informative edges. Second, true informative edges might have correlations to some other random edges, so removing edges randomly affects the capacity of the model to correctly learn important patterns in the graph.

*Are explanations obtained with graph explainability methods better than a random guess?* We observe that a random edge modification removes more informative edges than GradCAM, Integrated Gradient, Occlusion, RCExplainer, and PGExplainer. Therefore, those methods are not better than Random.

> GInX-Eval identifies how informative ground-truth explanations are for the model, thus assessing the agreement between model and human-based explanations, and draws attention to how much random explanations are meaningful to the model.

### 4.3.3 GINX-EVAL OF EXPLAINABILITY METHODS

*What fraction $t$ of removed edges should I fix to compare the GInX scores of multiple explainability methods?* Methods produce explanations of different sizes: some methods constrain their explanations to be sparse, while others assign importance weight to almost all edges in the graph. Figure 6 indicates the heterogeneity of masks generated by different explainability methods. While Truth, PGMExplainer, SubgraphX, and GraphCFE constrain their explanations to be sparse, the rest of the methods include most of the edges in the explanations, assigning a different importance weight to each edge.

The *critical threshold* $t_m^c$ of a method $m$ is the ratio of non-zero values in the masks. Beyond this critical threshold, we are not evaluating the method anymore but a random assignment of edge importance weights. Therefore, it is crucial to compare methods at a threshold $t$ smaller than the minimum of the methods' critical thresholds. To compare methods, we propose to define the dataset's *optimal threshold* $t^*$ such as $t^* = min_{m \in \mathcal{M}}\{t_m^c\}$, where $\mathcal{M}$ denotes the set of explainability meth-

ods. The optimal threshold corresponds to the threshold closest to the average mask sparsity of ground-truth explanations. In other words, we take as reference the size of ground-truth explanations as the optimal number of informative edges in the graph and compare methods at this sparsity threshold. We compute the optimal thresholds for the six datasets and report them in table 1. Only the BBBP dataset has no ground-truth explanation available so we set $t^* = 0.3$ to have human-intelligible sparse explanations.

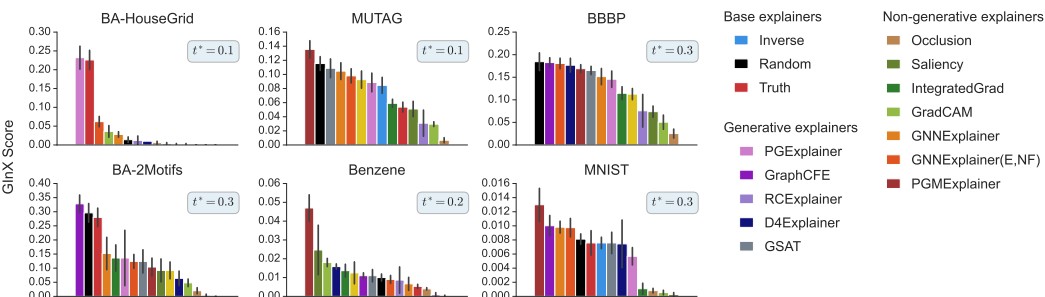

Figure 7: GInX scores at the optimal thresholds. For the BBBP dataset, we define an arbitrary optimal threshold $t = 0.3$. For the other datasets, the optimal threshold is estimated based on the explanatory mask sparsity generated by the Truth estimator.

Figure 7 displays the GInX scores of explainability methods at the optimal threshold defined for each dataset. Except for the Benzene dataset, we observe that gradient-based methods and Occlusion have the smallest GInX scores at the optimal thresholds. Gradient-based methods contain less informative edges than GNNExplainer, PGMExplainer, and generative methods. This contradicts observations made in figure 3 where gradient-based methods and Occlusion are always better than GNNExplainer and PGMExplainer. GInX-Eval unveils new insights on gradient-based methods that go against recent studies (Yuan et al., 2023; Agarwal et al., 2022). On the other hand, GN-NExplainer, PGMExplainer, GSAT, and D4Explainer have competitive performance with Random and Truth baselines. This proves that generative methods are not necessarily better at capturing meaningful information than non-generative methods.

> The GInX score at the optimal threshold helps filter out uninformative methods including gradient-based methods and Occlusion, and shows that methods can generate informative explanations independent of their generative nature.

### 4.3.4 HOMOPHILICRANK SCORE OF EXPLAINABILITY METHODS

We use the HomophilicRank score to evaluate the capacity of explainers to rank edges correctly according to their true informativeness for the model and identify redundant information. In figure 8, we observe that gradient-based methods and Occlusion are not good at correctly ordering edges by their importance. This is another reason why they should not be used to generate meaningful explanations. We also observe that RCExplainer and PGExplainer which perform well on the GInX score have a low edge ranking power, except for the BA-HouseGrid dataset. These two methods can capture the most informative edges but cannot decide what the relative importance of those important edges is. Finally, PGMExplainer, GNNexplainer, GraphCFE, and D4Explainer have both a high GInX score (see figure 5) and a high HomophilicRank score, making them the best choice for informative and edge-rank powerful explainers.

> With the HomophilicRank score, GInX-Eval indicates what method can better rank edges according to their informative power, assigning all correlated edges a similar importance. This helps to compare methods that already perform well on GInX score.

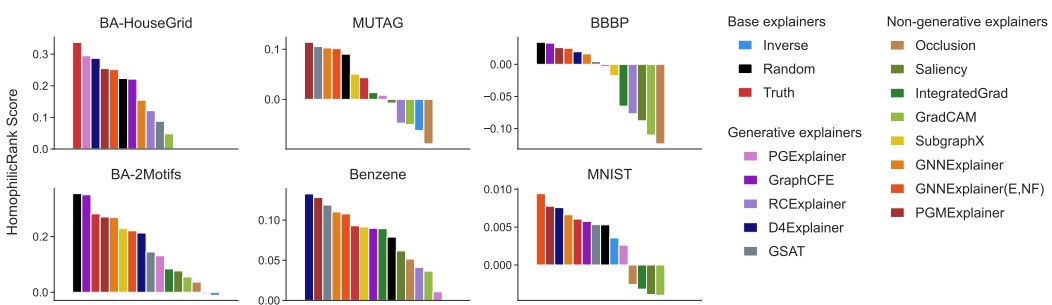

Figure 8: HomophilicRank scores of explainability methods.

# 5    DISCUSSION

This work discusses the pitfalls of faithfulness, one of the most popular metrics in xAI, and the problem of out-of-distribution explanations. Overcoming these limitations, our evaluation procedure GInX-Eval measures the informativeness of explainability methods and their ability to accurately rank edges by their importance for the GNN model. Observing the prediction change, GInX-Eval assesses the impact of removing the generated explanations from the graphs. It gets around the issue of OOD explanations by fine-tuning the GNN model. GInX-Eval is a useful tool to validate the quality of the provided ground-truth explanations. It also demonstrates the poor informativeness of gradient-based methods, contradicting results from recent studies (Yuan et al., 2023; Agarwal et al., 2022) and reproduced in this paper. Combining the GInX and HomophilicRank scores, we can filter out uninformative explainability methods and find the optimal ones. Because GInX-Eval relies on a fine-tuning strategy of pre-trained black-box models, our method can easily be used for models only accessible via API calls, including large language models. Due to the computation cost of re-training, GInX-Eval is not meant to be used systematically but is designed as a validation tool for new metrics. This work paves the way for developing approaches that conform with both human-and model-based explainability.

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
