# A    PREVIOUS EVALUATION FRAMEWORK

## A.1    FAITHFULNESS METRICS

**Remove**

$$fid{-}^{prob} = \frac{1}{N} \sum_{i=1}^{N} |p(f(G_i) = y_i) - p(f(h(G_i)) = y_i)|$$

$$fid{-}^{acc} = \frac{1}{N} \sum_{i=1}^{N} |\mathbb{1}(f(G_i) = y_i) - \mathbb{1}(f(h(G_i)) = y_i)|$$

**Keep**

$$fid{+}^{prob} = \frac{1}{N} \sum_{i=1}^{N} |p(f(G_i) = y_i) - p(f(G_i \setminus h(G_i)) = y_i)|$$

$$fid{+}^{acc} = \frac{1}{N} \sum_{i=1}^{N} |\mathbb{1}(f(G_i) = y_i) - \mathbb{1}(f(G_i \setminus h(G_i)) = y_i)|$$

Faithfulness or fidelity metrics (Yuan et al., 2023) evaluate the contribution of the produced explanations to the initial prediction, either by giving only the explanation as input to the model (fidelity-) or by removing it from the entire graph and re-run the model (fidelity+). The keep/removing is done according to hard or soft selection (See section 3.1) and the explanatory edge mask. The fidelity scores capture how well an explainable model reproduces the natural phenomenon. The fidelity is measured concerning the ground-truth label $y$. Equations A.1 detail the mathematical expressions of the different fidelity scores. We use the same notations as defined in section 3.1 where $f$ represents the GNN model and $h$ the explainability method. The fidelity scores (+/-) can be expressed either with probabilities ($fid_{+/-}^{prob}$) or indicator functions ($fid_{+/-}^{acc}$). While $fid_{+/-}^{prob}$ metrics are more appropriate for evaluating explanations in the context of regression tasks because they are only based on the predicted probabilities, $fid_{+/-}^{acc}$ metrics use the indicator function and are more suitable for classification problems.

## A.2    THE OUT-OF-DISTRBUTION PROBLEM

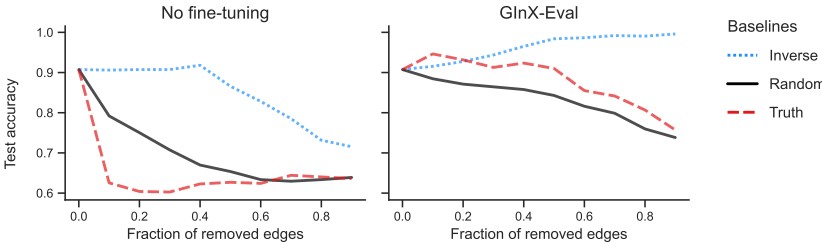

Figure 9: Comparison between not fine-tuning the GNN model and GInX-Eval on the MUTAG dataset when *hard* removing edges estimated informative by the three base estimators: Truth, Inverse, and Random. New graphs are obtained with the *hard* selection strategy, i.e., edges are strictly removed from the initial graph structure.

The Inverse estimator is a good indicator of the out-of-distribution problem existence. When you remove uninformative edges and the model is not fine-tuned, the test accuracy should stay constant. But in the case of out-of-distribution explanations, the model has never seen these instances and therefore is not able to predict the correct labels anymore. A robust model toward OOD is a model in which test accuracy does not degrade with the Inverse estimator, i.e., as we remove uninformative edges.

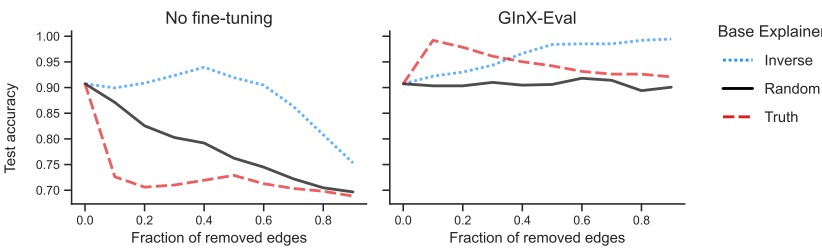

Figure 10: Comparison between not fine-tuning the GNN model and GInX-Eval on the MUTAG dataset when *soft* removing edges estimated informative by the three base estimators: Truth, Inverse, and Random. New graphs are obtained with the *soft* selection strategy, i.e. edges receive a zero weight if they must be removed.

In figures 9 and 10, we observe in both edge removal strategies a drop of model performance with the Inverse estimator when the model is not fine-tuned; but no degradation with GInX-Eval fine-tuning. Fine-tuning the GNN model at each degradation level overcomes the OOD problem and enables a robust evaluation.

**Remark** The degradation of the GNN test accuracy is smaller with fine-tuning than without. For MUTAG dataset we observe in figure 9 a drop in performance for the Truth estimator of 0.18 with GInX-Eval versus 0.25 with no fine-tuning at 90% edge removal. The model is remarkably robust to edge modification. After removing a large portion of all edges, the model still learns to make correct predictions and retain almost most of the original predictive power. For instance, for the MUTAG dataset, a random modification of 50% of the edges degrades the accuracy only to 85% accuracy. The model can extract meaningful representations from a small amount of remaining edges, which suggests that many edges are likely redundant or correlated.

## A.3 HARD VERSUS SOFT SELECTION

In this section, we explore the effect of the selection strategy on GInX-Eval. With hard edge removal, the drop in test accuracy is significant, in the order of $10^{-1}$. For the soft edge removal, the drop is very small in the order of $10^{-2}$ or $10^{-3}$ for some datasets. In figure 10, we observe for the MUTAG dataset that, after removing 90% of the edges and when the model is fine-tuned, the accuracy drops by 10% for the Truth estimator in the case of soft selection, against 28% with the hard selection on figure 9. This suggests that removing informative edges by setting their weights to zeros does not prevent the model from learning from the masked information. For soft selection, the removed informative edges are not fully ignored. On the contrary, with hard selection, the model has no way of capturing the removed information since edges are fully removed from the structure. For this reason, we favor hard edge selection, even though similar effects are observed at different scales and similar conclusions are drawn with both types of selection strategies (See Appendix C.2).

## A.4 MESSAGE PASSING WITH EDGE WEIGHTS

To understand why the soft selection enables the model to capture masked information and be robust to edge removal, we explain here how the soft selection affects the GNN training. In a weighted graph, each edge is associated with a semantically meaningful scalar weight. For soft explanations, for instance, the edge weights are importance scores. Graph neural networks integrate the graph topology information in forward computation by the message-passing mechanism. Edge weights modify the message from node $i$ to node $j$ in different ways according to the type of convolutional layer. For GAT convolutional layers, the attention scores are updated with additional edge-level attention coefficients $\alpha$ that correspond to the edge weights, so that $\alpha = \alpha_{\text{node}} + \alpha_{\text{edge}}$. For GINE convolutional layers, the message is the node vector $x_j$ followed by a ReLU activation. To account for edge importance, the message is modified by adding to the node vector the transformed edge weight $Lin(w_{ij})$. By modifying each message with its corresponding edge weight, the model keeps the whole graph connectivity but learns the importance of the node connections.

| | **Real-world** | | | | **Synthetic** | |
|---|---|---|---|---|---|---|
| **Datasets** | MUTAG | BBBP | Benzene | MNISTbin | BA-2Motifs | BA-HouseGrid |
| # Graphs | 2,951 | 2,039 | 12,000 | 14,780 | 1,000 | 2,000 |
| Avg # Nodes | 30 | 24 | 20.6 | 63 | 25 | 101 |
| Avg # Edges | 61 | 51.9 | 43.7 | 486.6 | 51 | 817 |
| # Node features | 14 | 9 | 14 | 5 | 1 | 1 |
| # Edge features | 1 | 3 | 5 | 1 | 1 | 1 |
| # Classes | 2 | 2 | 2 | 2 | 2 | 2 |
| GT Explanation | $NH_2$, $NO_2$ | - | Benzene ring | Figure pixels | House, cycle | House, grid |

Table 2: Datasets for graph classification

# B EXPERIMENTAL DETAILS

## B.1 DATASETS

We evaluate the explainability methods on both synthetic and real-world datasets used for graph classification tasks.

*BA-2Motifs* (Luo et al., 2020) is a synthetic dataset with binary graph labels. The house motif and the cycle motif give class labels and thus are regarded as ground-truth explanations for the two classes.

*BA-HouseGrid* is our new synthetic dataset where we attach house or grid motifs to a Barabási base structure. We choose two distinct motifs, house and grid, where one is not the subgraph of the other, unlike BA-2Motifs with the cycle that is contained in the house motif. The datasets contain 2,000 graphs with balanced BA-Grid and BA-House graphs. The base Barabási graphs contain 80 nodes, and the number of motifs varies between 2 and 5 per graph.

*MUTAG* is a collection of ∼3,000 nitroaromatic compounds and it includes binary labels on their mutagenicity on Salmonella typhimurium. The chemical fragments -NO2 and -NH2 in mutagen graphs are labeled as ground-truth explanations (Luo et al., 2020).

*BBBP* includes binary labels for over 2,000 compounds on their permeability properties (Wu et al., 2018). The task is to predict the target molecular properties. In molecular datasets, node features encode the atom type and edge features encode the type of bonds that connect atoms.

*Benzene* contains 12,000 molecular graphs extracted from the ZINC15 (Sterling & Irwin, 2015) database and labeled into two classes where the task is to identify whether a given molecule has a benzene ring or not. The ground-truth explanations are the nodes (atoms) comprising the benzene rings, and in the case of multiple benzenes, each of these benzene rings forms an explanation.

*MNISTbin* contains graphs that are converted from images in MNISTbin (LeCun et al., 1998) using superpixels. In these graphs, the nodes represent the superpixels, and the edges are determined by the spatial proximity between the superpixels. The coordinates and intensity of the corresponding superpixel construct the node features. We reduce the MNISTbin graph dataset for binary classification by selecting only the input of classes 0 and 1.

## B.2 GNN PRE-TRAINING

We test three GNN models: GCN Kipf & Welling (2017), GIN Hu et al. (2020) and GAT Veličković et al. (2018) because they score high on the selected real-world datasets, with a reasonable training time and fast convergence. The network structure of the GNN models for graph classification is a series of 3 layers with ReLU activation, followed by a max pooling layer to get graph representations before the final fully connected layer. We split the train/validation/test with 80/10/10% for all datasets and adopt the Adam optimizer with an initial learning rate of 0.001. Each model is initially

trained for 200 epochs with an early stop. Each training is always repeated on five different seeds. ==Unlike the GCN model, GAT and GIN models can take edge features as additional input data. For this reason, BBBP and Benzene datasets can not be tested with the GCN model.==

We report in Table 3 ==the test accuracy of GCN, GAT, and GIN models for all six datasets. We report the average and standard error of the experiments run on five different seeds. We observe high test accuracy, i.e., $> 0.8$, for all datasets for the GIN model and all real-world datasets for the GAT and GCN models. Only for BA-2Motifs and BA-HouseGrid is the test accuracy of the GAT and GCN models not good enough to be tested in the explainability analysis and evaluated with GInX-Eval.==

|  | **BA-2Motifs** | **BA-HouseGrid** | **BBBP** | **Benzene** | **MNISTbin** | **MUTAG** |
|---|---|---|---|---|---|---|
| GCN | $0.408_{\pm 0.048}$ | $0.512_{\pm 0.064}$ | - | - | $0.994_{\pm 0.003}$ | $0.889_{\pm 0.021}$ |
| GAT | $0.428_{\pm 0.068}$ | $0.512_{\pm 0.034}$ | $0.839_{\pm 0.024}$ | $0.917_{\pm 0.044}$ | $0.992_{\pm 0.001}$ | $0.905_{\pm 0.022}$ |
| GIN | $0.988_{\pm 0.016}$ | $1.00_{\pm 0.000}$ | $0.835_{\pm 0.035}$ | $0.999_{\pm 0.001}$ | $0.995_{\pm 0.002}$ | $0.907_{\pm 0.016}$ |

Table 3: ==Test accuracy of pre-trained GCN, GAT and GIN== graph neural networks for the six datasets on graph classification task.

### B.3 GInX-Eval algorithm

---

**Algorithm 1** GInX-Eval, an in-distribution evaluation procedure for explainability methods

---

**Input**: $h$: explainer function, $f$: pre-trained GNN model, $\mathcal{G}$: set of graph instances
**Output**: $L$: list of GInX scores at different degradation levels

1: **function** GInX-Eval $(h, f, \mathcal{G})$
2:   Initialize an array to store evaluation results at different degradation levels
3:   $L \leftarrow []$
4:   **for** $t$ in $[0.1, 0.2, ..., 1]$ with a step size of 0.1 **do**
5:     Gather explanations produced by the explainer for all graph instances
6:     $\mathcal{G}^* \leftarrow h(\mathcal{G})$
7:     Rank the explanatory edges based on their weights
8:     $ranked\_edges \leftarrow \text{sort\_edges\_by\_weight}(\mathcal{G}^*)$
9:     Calculate the number of edges to remove based on the degradation level
10:    $N \leftarrow t \cdot |\mathcal{G}|$
11:    Remove the top $t$ fraction of edges from the input graphs
12:    $\mathcal{G}_{\text{train}}, \mathcal{G}_{\text{test}} \leftarrow \text{generate\_train\_test\_datasets}(\mathcal{G}, ranked\_edges, N)$
13:    Fine-tune the initial GNN model on the new train dataset
14:    $f \leftarrow \text{fine\_tune\_GNN}(f, \mathcal{G}_{\text{train}})$
15:    Evaluate the fine-tuned model on the new test data
16:    $TestAcc \leftarrow \text{evaluate\_model}(f, \mathcal{G}_{\text{test}})$
17:    Calculate the GInX score
18:    $GInX \leftarrow 1 - TestAcc$
19:    Store the GInX score for this degradation level
20:    $L.\text{append}(GInX)$
21: **end for**
22: **return** L
23: **end function**

---

### B.4 GInX-Eval computation time

Here we report the time for evaluating with GInX-Eval. GInX-Eval requires 10 GNN fine-tunings if we vary the threshold $t$ from $[0, 0.1, ..., 0.9]$. In table 4, we report the average GNN training time for GIN and GAT models for each dataset. The fine-tuned GNN models are trained in the same setting as the initial GNN model: the pre-trained model is fine-tuned for 200 epochs with an early stop.

Each training is always repeated on five different seeds. In the case of the six datasets tested in this paper, the re-training strategy has a low computational burden. In the case of large-scale datasets, selecting just a representative train/test subset can also speed up GInX-Eval.

|      | BA-2Motifs | BA-HouseGrid | BBBP | Benzene | MNISTbin | MUTAG |
|------|-----------|-------------|------|---------|----------|-------|
| GAT  | 28.1      | 63.4        | 39.4 | 209.1   | 290.7    | 79.0  |
| GIN  | 23.9      | 53.5        | 28.8 | 159.0   | 225.1    | 66.8  |

Table 4: Fine-tuning times (s) of GIN and GAT graph neural networks for the six datasets on graph classification task.

### B.5 GENERATIVE EXPLAINABILITY METHODS

Non-generative explainability methods like gradient-based or perturbation-based methods optimize individual explanations for a given instance. They lack a global understanding of the whole dataset as well as the ability to generalize to new unseen instances. To tackle this problem, non-generative methods have been developed. They learn the initial data distribution before generating individual explanations. Therefore, generative methods learn the underlying distributions of the explanatory graphs across the entire dataset, providing a more holistic approach to GNN explanations. The recent study of Chen et al. shows the superiority of generative methods and in particular concerning their generalization capacity and faster inference time. GInX-Eval also proves that generative methods are the only type of methods that can do better than a random guess, and this is consistent across datasets and GNN models.

### B.6 EXPLAINER HYPERPARAMETER SELECTION

Some explainability methods require hyperparameter tuning. The final parameters we decided on for GNNExplainer and PGExplainer two techniques that are highly dependent on parameter selection are listed in Tables 5 and 6.

Table 5: Final hyperparameters for GNNExplainer for edges and node features

| Graph Entity | Edge | Node Feat |
|--------------|------|-----------|
| Size regularization coeff | 0.005 | 1 |
| Entropy coeff | 1 | 0.1 |
| Reduction function | sum | mean |

Table 6: Final hyperparameters for PGExplainer

| | |
|--------------------------|--------|
| Size regularization coeff | 0.01 |
| Entropy coeff | $5e-4$ |
| Initial Temperature | 5 |
| Final Temperature | 1 |

### B.7 CODE IMPLEMENTATION

Our code is implemented using torch-geometric 2.3.0 (Fey & Lenssen, 2019) and Torch 1.9.1 with CUDA version 11.1 (Collobert et al., 2011; NVIDIA Corporation, 2023). The generation of edge masks by the explainability methods and the training of the graph neural networks at each removal threshold are performed on a Linux machine with 1 GPUs NVIDIA RTX A6000 with 10 GB RAM memory. The code is available at `https://anonymous.4open.science/r/GInX-Eval`.

## C ADDITIONAL RESULTS

### C.1 MORE EXPLAINABILITY METHODS

In Figure 11, we provide the GInX scores for all the tested methods: three baselines, eight non-generative methods, and five generative ones.

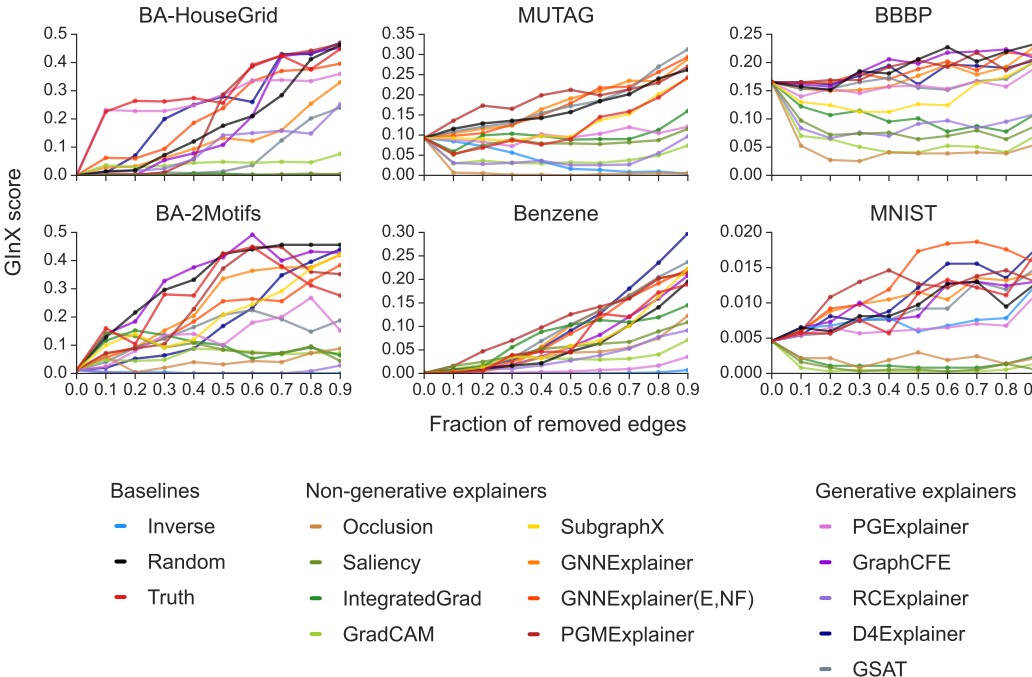

Figure 11: GInX scores of a fine-tuned GIN model on graphs with increasing fraction of removed edges. The removed edges are the most important based on explainability methods, and new input graphs are obtained with the *hard* selection, i.e. explanatory edges are strictly removed from the graph so that the number of edges and nodes is reduced.

*Why is the GInX score not increasing linearly?* We found two possible reasons for the nonlinear rise of the GInX score when removing important edges. One reason is that edges might be redundant: some correlations exist between them and their effect is neutralized only if all are removed. Another reason is that important edges are not always correctly ordered or have no order of importance, e.g. Truth assigns a weight of 1 to all ground-truth edges. In these cases, the true least important edge among important edges according to the estimator might be removed first, not degrading the performance of the model that is still trained with the most informative edges.

## C.2 GInX-Eval with Soft Selection

In the paper, we describe the different selection strategies to remove edges from a graph (see section 3.1) and explain our preference for the *hard* selection to convey the results in section 4.3. Here, we add the results of GInX-Eval for soft selection. The experimental setting is the same. We display the results for the four real-world datasets.

Compared to the results with hard selection, the GInX score increase here is way smaller, in the order of 0.01, as mentioned in the paper. Instead of observing a sharp rise in the GInX score for the best methods, we observe that the GInX score stays constant. The impact of masking informative edges does not prevent the model from learning because the model can still pass the message on the node level. For more details, we refer the reader to the discussion in Appendix A.4. For the worse methods, the GInX score decreases instead of staying constant. Removing uninformative edges by setting their weight to zero brings the model to better learn the actual class of the graphs. Only for Benzene, the Random and Truth baselines as well as most of the generative methods produce a $\sim 0.07$ rise of the GInX score.

Results on the MNISTbin dataset cannot be interpreted since the increase of the GInX score is of the order of $10^{-3}$ and Truth and Inverse estimators have the same behavior. For the three other datasets Benzene, MUTAG, and BBBP, figure 12 confirms the superiority of the non-generative methods GN-NExplainer and PGMExplainer, and the generative methods GSAT, GraphCFE, and D4Explainer.

Once again GradCAM, Occlusion, and the gradient-based methods Saliency, Integrated Gradient, and GradCAM capture the less informative edges, contradicting the previous studies on explainability for GNN (Yuan et al., 2023; Agarwal et al., 2022).

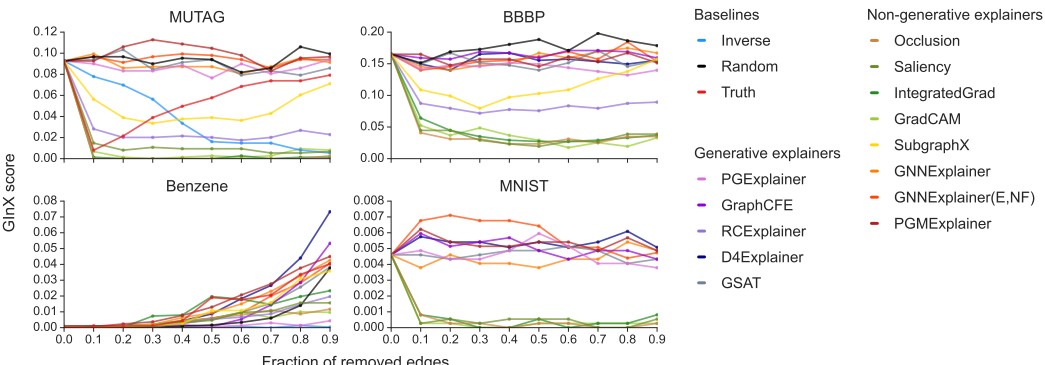

Figure 12: GInX scores of a fine-tuned GIN model on graphs with increasing fraction of removed edges. The removed edges are the most important based on explainability methods, and new input graphs are obtained with the *soft* selection, i.e. explanatory edges are masked so that their respective weight is put to zero.

## C.3   More GNN models

This section is dedicated to GInX-Eval with Graph Attention Networks (GAT) and Graph Convolutional Networks (GCN). Because GAT and GCN reach low accuracy with the two synthetic graph datasets, we only show the results for the real-world datasets. GCN model does not take into account edge features. For this reason, we can only run it for MUTAG and MNISTbin datasets. Figure 13 and 14 show the GInX-scores of GAT and GCN models fine-tuned on new graphs obtained after a hard edge removal strategy.

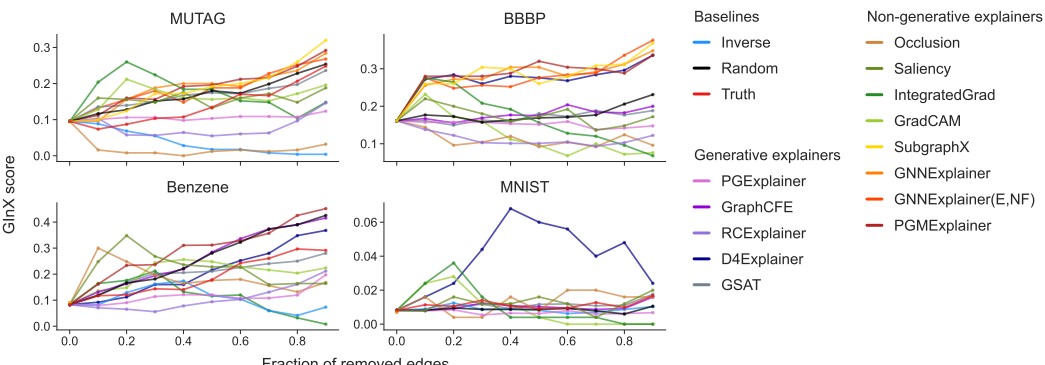

Figure 13: GInX scores of a fine-tuned GAT model on graphs with increasing fraction of removed edges. The removed edges are the most important based on explainability methods, and new input graphs are obtained with the *hard* selection, i.e. explanatory edges are strictly removed from the graph so that the number of edges and nodes is reduced.

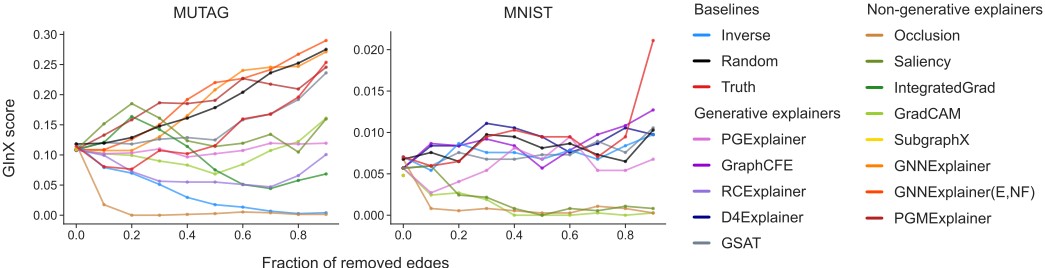

Figure 14: GInX scores of a fine-tuned GCN model on graphs with increasing fraction of removed edges. The removed edges are the most important based on explainability methods, and new input graphs are obtained with the *soft* selection, i.e. explanatory edges are masked so that their respective weight is put to zero.