# OpenReview forum: "GInX-Eval: Towards In-Distribution Evaluation of Graph Neural Networks Explanations"
_ICLR.cc/2024/Conference — Submitted to ICLR 2024_

### Official Review · Reviewer_DCst · 2023-10-27

**Soundness:** 3 good
**Presentation:** 4 excellent
**Contribution:** 2 fair
**Rating:** 6
**Confidence:** 4

**Summary:**

This paper focuses on the problem of out-of-distribution explanations, which means in the explainability tasks of graph neural networks, the highlighted explanation subgraph’s distribution differs from the training data. The existing evaluation metrics such as faithfulness or fidelity score couldn’t evaluate the explanation well due to the OOD issue. The author proposed GInX-Eval to better evaluate the explainers by retraining the GNN model and showed its great evaluation performance.

**Strengths:**

1. This paper has a great impact on the domain of the XAIG. It addresses a common concern about how the OOD problem affect the performance of the commonly used faithfulness metric.
2. Figure 2 shows the effectiveness of the proposed methods greatly.
3. The claims in section 4 are easy to follow and good to refer to.
4. This paper has a good presentation and is easy to follow.
5. The experiments are solid and sufficient.

**Weaknesses:**

1. GInX-Eval has to treat the pre-trained model as a white box because it needs to retrain the model during the whole procedure. However, the pre-trained to-be-explained model is not always a white box, especially in real-life applications. The training dataset may not be able to be accessed, or the training cost is high, even the model itself may be not accessible. So, this approach is not easy to apply.
2. The methodology itself is not novel enough. Remove and retrain is not new in the machine learning community, eg: “Sara Hooker, Dumitru Erhan, Pieter-Jan Kindermans, Been Kim, 2019, A Benchmark for Interpretability Methods in Deep Neural Networks”.
3. The contributions are over-claimed. Some previous work have also addressed the OOD problem, eg:

[1] “Junfeng Fang, Xiang Wang, An Zhang, Zemin Liu, Xiangnan He, and Tat-Seng Chua. 2023. Cooperative Explanations of Graph Neural Networks. In Proceedings of the Sixteenth ACM International Conference on Web Search and Data Mining (WSDM '23). Association for Computing Machinery, New York, NY, USA, 616–624. https://doi.org/10.1145/3539597.3570378"

[2] “J Fang, W Liu, A Zhang, X Wang, X He, K Wang, TS Chua. On Regularization for Explaining Graph Neural Networks: An Information Theory Perspective ”

[3] “Jiaxing Zhang, Dongsheng Luo, Hua Wei. 2023. MixupExplainer: Generalizing Explanations for Graph Neural Networks with Data Augmentation. SIGKDD’23”

[4] "Ying-Xin Wu, Xiang Wang, An Zhang, Xia Hu, Fuli Feng, Xiangnan He and Tat-Seng Chua, 2022, Deconfounding to Explanation Evaluation in Graph Neural Networks.”

**Questions:**

Comments:
1. “The highlighted explanation subgraph’s distribution is different from the training data.” Why is different and what’s the nature where this difference comes from?
2. Why did the explanations’ distribution shift to a better side but not a worse side? For example: the prediction label is 50. A good explanation prediction should be 50. A bad explanation prediction should be 20. However, due to the  OOD, the explanation prediction shifts. Why a bad prediction would shift from 20 to 45 and cause an incorrect high faithfulness score, instead of shifts from 20 to 5? As it’s claimed: “However, this edge masking strategy creates Out-Of-Distribution (OOD) graph inputs, so it is unclear if a high faithfulness score comes from the fact that the edge is important or from the distribution shift induced by the edge removal (section 1 paragraph 1)”.
3. There are two removal strategies: “hard” and “soft” removal strategies. I wonder is there any difference between them toward the GNN output? If the outputs f(G_e_hard) and f(G_e_soft) are different, what’s the reason for that?
4. There should be many hyper-parameters to tune for the evaluated explainer methods, eg: size regularization and temperature in GNNExplainer/PGExplainer. How do you set them and have you tuned them to the best? It would be good to include these details in the main text or supplementary and motion them in the main text since this paper emphasizes on the experiments.
5. In Figure 1, what’s the random seed for the random baseline, and how many times the experiments are repeated? For AUC evaluation, how do you compute the AUC score? Specifically, for other explainers, we could have an edge weight vector as the explanation and compute the AUC with the ground truth. But for a random baseline, how to decide the weight of the edge?
6. The GInX-Eval is computined via retraining, and finally evaluating the quality of the explanation of the original on the original pretrained GNN model. However, the GNN behavior would change during retraining. For example: GNN model f_a is trained on the complete training dataset, it could predict the classification according to the explanation sub-graph. But GNN model f_a is trained on the training dataset which frop 50% edges in each graph. If the explanation sub-graphs are already dropped, how could f_b predicts the graphs into correct classifications? Would the behavior of the retrained GNN models change and how would it affect the accuracy evaluation? Thus, the experiments are not fully convincing. It would be good to make some clarify.


Typos:
1. In section 2, “Solving the OOD problem” should be “Solving the OOD Problem”.
2. In section 3, “Edge removal strategies”, “Prior work” should be “Edge Removal Strategies” and “Prior Work” to be consistent with “Out-Of-Distribution Explanations”
3. In section 4, “Experimental setting” should be “Experimental Setting”.
4. In the “Experimental setting” section, “We test two …, because they score high on …”: should it be “because their scores are high on…”?

---

> ### Author Response · Authors · 2023-11-12
> **Answer to Reviewer DCst**
>
> We thank Reviewer DCst for his/her insightful comments.
>
> One major comment of the reviewer concerns the limited applicability of the method to white-box models. Now we propose a fine-tuning strategy that enables both to overcome the OOD problem and to be generalizable to any black-box models only accessible through API calls. See general comments to all reviewers.
>
> Although the methodology is not novel, the whole purpose of this paper is to bring a new evaluation tool to the xAI and GNN community to avoid common mistakes and get meaningful insights into the true informative power of explainability methods as well as validate ground-truth explanations, something that is currently missing in the community!
>
> This paper doesn’t claim to solve the OOD problem, it shows first the limitation of a popular evaluation metric in xAI that is not robust to the OOD problem as well as the consequences (observation1,2,3) and proposes a novel evaluation strategy that overcomes this.
>
> 1. *“The highlighted explanation subgraph’s distribution is different from the training data.” Why is different and what’s the nature where this difference comes from?*
>
> We have added Figure 2 to illustrate the OOD problem.
>
> 2. *Why did the explanations’ distribution shift to a better side but not a worse side? ... (section 1 paragraph 1)”*.
>
> Since explanations are out-of-distribution, there is no general rule beyond the observed faithfulness scores. The idea here is that the GNN is fooled and therefore has not the capacity to correctly assess the importance of the remaining edges. Figure 2 very well illustrates that explanations are closer to non-toxic molecules and will therefore be misclassified even though they are good explanations for toxic molecules (if we look at the GT or GinX-Eval results).
>
> 3. *There are two removal strategies: “hard” and “soft” removal strategies. I wonder is there any difference between them toward the GNN output? If the outputs f(G_e_hard) and f(G_e_soft) are different, what’s the reason for that?*
>
> We refer the reviewer to the Appendix where he will find a more detailed analysis of the differences in the GNN output between hard and soft edge removal strategies. In Appendix A.3  we analyze the differences in the GNN output between hard and soft edge removal and in Appendix C.2 we also provide the results for the soft edge removal strategy and describe the similarities and differences with the results with hard selection.
>
> 4. *There should be many hyper-parameters to tune ...*
>
> We have included the details of hyperparameters for GNNEXplainer and PGExplainer in the Appendix. We fully agree with the reviewer that PGExplainer is extremely sensitive to the choice of hyper-parameters and our final choice is the result of hyper-parameter tuning.
>
> 5. *In Figure 1, what’s the random seed for the random baseline...*
>
> For each experiment (the random baseline included), we have run the experiments on five different seeds (0,1,2,3,4). The random baseline estimates the edge importance sampling from uniform distribution values between 0 and 1. The weight of the edge for the random explanation is therefore a random value between 0 and 1. To compute the AUC score, we simply compare this random edge importance assignment to the ground-truth one. We have added a sentence in section 4.1 to clarify the AUC computation. “The AUC score is computed between the explanatory weighted edge mask and the ground-truth edge mask with binary values in {0,1}.”
>
> 6. *The GInX-Eval is computined via retraining, and finally evaluating the quality of the explanation of the original on the original pretrained GNN model. However, the GNN behavior... clarify.*
>
> As mentioned in the general comment to all reviewers, we now propose a fine-tuning strategy. When removing t% of the informative edges, f_b(fine-tuned on the reduced train dataset) will not find the important information in the reduced test set - therefore we expect f_b to make wrong predictions and the test accuracy to drop and the GinX score to increase.

---

### Official Review · Reviewer_qa24 · 2023-10-28

**Soundness:** 3 good
**Presentation:** 3 good
**Contribution:** 3 good
**Rating:** 6
**Confidence:** 3

**Summary:**

This paper discusses the evaluation method of explanatory techniques for GNNs. It argues that the faithfulness measure commonly used in the GNN explainability research area suffers from the out-of-distribution (OOD) problem where removing uninformative edges can decrease accuracy because they lead to the OOD. To tackle this problem it proposes GInX-Eval that evaluates explanatory techniques according to the decrease in test accuracy on GNNs retrained by using training data in which the highly-ranked edges are subtracted. It empirically shows that the faithfulness score is inconsistence with accuracy and decreases by removing even the uninformative edges, whereas GInX-Eval does not suffer from removing the uninformative edges. The results based on GInX-Eval indicate that some explanatory techniques like gradient-based methods have not good performance whereas others such as GNNExplainer and D4Explainer can provide good explanations of GNN predictions, which are consistent with the results of previous works.

**Strengths:**

Overall, this paper is well-organized and clearly written. This paper clearly proves the problem of the faithfulness measure widely used in the GNN explainability research community by using carefully designed experiments. The proposed measure, GInX-Eval, can overcome the OOD problem from which the faithfulness measure suffers, by observing the test accuracy on GNNs retrained by using the training data. The evaluation based on GInX-Eval is consistent with the results of the previous works.

**Weaknesses:**

Though GInX-Eval is designed so that it can be applied to graph data and it provides good contributions to the graph learning research area, the idea of evaluating explanatory techniques by retraining the prediction methods has already been proposed in previous works such as Hooker et al (2018).

Additionally, there are several drawbacks to readability:
- In 3.3.1 GINX SCORE, the description of "top-k edges" is confusing because t is already used as the fraction of the ordered edge set.
- In equation 3, the superscript for G\G_e^t is used without explanation despite the superscript is not used in equation 2.
- It is very hard for readers to distinguish different colors used in Figures. Some efforts are required for readability such as using different marks.
- Several references such as Faber et al, Hooker et al, Hsieh et al, and Hu et al lack names of conferences or years of publishing.

**Questions:**

What is the difficulty of applying the idea of retraining to the evaluation of explanatory techniques for GNNs compared to those for CNNs?

---

> ### Author Response · Authors · 2023-11-12
> **Answer to Reviewer qa24**
>
> We thank Reviewer qa24 for his/her insightful comments.
>
> *What is the difficulty of applying the idea of retraining to the evaluation of explanatory techniques for GNNs compared to those for CNNs?*
>
> Thanks for the question. Theoretically, GInX-Eval should be easily transferable to CNN models as well. However, the difficulty with CNNs comes from the nature of the inputs themselves, i.e. images. The challenges arise when removing pixels from images. The removal strategy of pixels when using GInX-Eval with CNNs has to be well justified. Two main challenges should be addressed:
>
> - As mentioned by Hsieh et al. [1], assigning features with the baseline value might lead to a bias toward features that are far from the baseline. For instance, with images the baseline is black color. If we set the pixels to black in RGB images, this introduces a bias favoring bright pixels. Therefore, CNN fine-tuned on those modified images will often omit important dark objects.
> - In addition, removing pixels by setting their value to zero (black pixel) might lead to the CNN confusion between no signal composing the black background and pertinent negative pixels, which are relevant values that are dismissed [2].
>
> Therefore, applying GInX-Eval for CNN models requires defining an appropriate pixel removal strategy to overcome those two difficulties.
>
> [1] Hsieh, Cheng-Yu, Chih-Kuan Yeh, Xuanqing Liu, Pradeep Ravikumar, Seungyeon Kim, Sanjiv Kumar, and Cho-Jui Hsieh. n.d. “Evaluations and Methods for Explanation through Robustness Analysis.” https://openreview.net/pdf?id=4dXmpCDGNp7.
>
> [2] Dhurandhar, Amit, Pin-Yu Chen, Ronny Luss, Chun-Chen Tu, Paishun Ting, Karthikeyan Shanmugam, and Payel Das. 2018. “Explanations Based on the Missing: Towards Contrastive Explanations with Pertinent Negatives.” arXiv [cs.AI]. arXiv. https://proceedings.neurips.cc/paper_files/paper/2018/file/c5ff2543b53f4cc0ad3819a36752467b-Paper.pdf.
>
>
> We have addressed the concerns of the reviewer in the updated version of the paper. As mentioned in the general comment for all reviewers, we have modified section 3 with more consistent mathematical notations and this also modified Equation 3. In 3.3.1, we have corrected the “top-k edges” and clarified that t is the fraction/ratio of edges. We have also completed the incomplete references when papers were published at conferences and added the year for all of them. Concerning the colors, we have chosen colorings to reflect the similarity of methods (green colors for gradient-based methods, warm colors for non-genative ones, and cold colors for generative ones).
>
> As mentioned in the general comment to all reviewers, we now proposed a fine-tuning strategy rather than re-training from scratch the GNN model. Therefore, the GInX-Eval method can now be applied to any black-box models, including language models and CNNs.

---

> > ### Comment · Reviewer_qa24 · 2023-11-18
> >
> > Thank you for your detailed comments. I understood that there are different difficulties between the evaluation of the CNN explanation and that of the GNN explanation. Even taking into consideration, the strengths and weaknesses of this paper have not changed so much for me, thus I keep the rating.

---

### Official Review · Reviewer_EeHX · 2023-11-01

**Soundness:** 2 fair
**Presentation:** 2 fair
**Contribution:** 2 fair
**Rating:** 5
**Confidence:** 4

**Summary:**

This paper proposes a new evaluation procedure for graph neural network (GNN) explanations called GInX-Eval. The authors argue that current evaluation metrics have limitations, particularly in evaluating out-of-distribution explanations. GInX-Eval addresses this issue by measuring the informativeness of removed edges and the correctness of explanatory edge ordering. The authors also introduce a new dataset for evaluating GNN explanations and demonstrate the effectiveness of GInX-Eval through experiments on this dataset. Overall, the paper's contributions include a new evaluation metric for GNN explanations, a new dataset for evaluation, and experimental results demonstrating the effectiveness of GInX-Eval.

**Strengths:**

- Proposes a novel evaluation metric, GInX-Eval, that measures the informativeness of removed edges and the correctness of explanatory edge ordering.
- Addresses an important issue in current evaluation metrics, namely the problem of out-of-distribution explanations.
- Clear and well-organized writing that makes it easy to follow the authors' arguments and contributions.

**Weaknesses:**

1. Certain aspects of the design are not intuitively clear. Specifically, the rationale behind Equation 4 is not well-explained. Elaborating on the underlying intuition would aid in understanding its relevance and function within the model.
2. The terms "hard selection" and "soft selection" are used without formal definitions. Providing precise mathematical formulas for these concepts would clarify their meaning and implementation in the context of the proposed method.
3. A major concern with GINX-EVAL is that it necessitates the re-training of the evaluated model. This process alters the original model, potentially leading to explanations that do not accurately reflect the model's decision-making process in its original state.
4. The utility of edge ranking as a metric is questionable. It assumes that the importance of individual edges correlates directly with subgraph importance, an assumption that may not hold true in all cases. Further justification or alternative metrics should be considered.
5. The range of GNN backbones tested is somewhat limited. Incorporating more diverse architectures, such as GCN, would provide a more comprehensive evaluation of the proposed method's effectiveness across different models.

In summary, while the paper introduces an intriguing approach for GNN evaluation, there are several areas where clarity and methodological rigor could be improved. Addressing these concerns would significantly enhance the paper's contribution and applicability.

**Questions:**

In weakness

---

> ### Author Response · Authors · 2023-11-12
> **Answer to Reviewer EeHX**
>
> We thank Reviewer EeHX for his/her insightful comments and address them one by one.
>
> 1. *Certain aspects of the design are not intuitively clear. Specifically, the rationale behind Equation 4 is not well-explained. Elaborating on the underlying intuition would aid in understanding its relevance and function within the model.*
>
> As explained in comment 5. below, we have re-formulated the EdgeRank score to the HomophilicRank score and elaborated more on the meaning of this score in the new version of the paper.
>
> 2. *The terms "hard selection" and "soft selection" are used without formal definitions. Providing precise mathematical formulas for these concepts would clarify their meaning and implementation in the context of the proposed method.*
>
> We have added a mathematical definition of the hard and soft edge selection functions using the notations of the paper to bring more clarity to the paper.
>
> 3. *A major concern with GINX-EVAL is that it necessitates the re-training of the evaluated model. This process alters the original model, potentially leading to explanations that do not accurately reflect the model's decision-making process in its original state.*
>
> We have addressed this major issue by proposing instead a fine-tuning strategy that enables both to overcome the OOD problem and to have an evaluation method that can be widely used for black-box models.
>
> 4. *The utility of edge ranking as a metric is questionable. It assumes that the importance of individual edges correlates directly with subgraph importance, an assumption that may not hold true in all cases. Further justification or alternative metrics should be considered.*
>
> The EdgeRank score does not evaluate the importance of individual edges but the importance of the first fraction $t\in[0.1,0.2,…0.9]$ of edges. We agree with the reviewer that the 10% most important edges might be correlated with the 10-20% edges, leading to an increase in the GInX score only after removing the full 20% edges. However, observing the size of ground-truth explanations for the 6 datasets (between 10 and 30%), we expect the sharpest drop in test accuracy in the first 3 removal steps (t=0.1, t=0.2, t=0.3). Therefore assigning more importance to these first steps makes sense. Now, it is true that within the 3 first steps, important edges might be correlated.
>
> We follow the reviewer’s suggestion and re-formulate the EdgeRank score to the HomophilicRank score. The HomophilicRank score measures the capacity of a method to rank edges by their correct importance ordering while assigning similar importance weights to correlated edges. This score penalizes methods that for instance only discover a subset of important edges and do not account for the edges correlated to those. The HomophilicRank score favors methods that treat correlated edges on equal footing. It measures the capacity to uniformly assign importance to redundant information rather than only putting importance to a single representative edge and none to the correlated edges.
>
> 5. *The range of GNN backbones tested is somewhat limited. Incorporating more diverse architectures, such as GCN, would provide a more comprehensive evaluation of the proposed method's effectiveness across different models.*
>
> We followed the reviewer’s suggestion and added the GCN model. Since the GCN does not take edge features into account we only run GInX-Eval with GCN on the four datasets: BA-2Motifs, BA-HouseGrid, MUTAG, and MNIST_BIN where edge features could be removed without or with small loss of information.

---

> > ### Comment · Reviewer_EeHX · 2023-11-22
> > **Response**
> >
> > Thank you for your feedback. After considering the revised strengths and weaknesses of this paper, I acknowledge its improvements but believe it is not yet ready for publication. Consequently, I have increased its score to 5.

---

### Author Response · Authors · 2023-11-12
**General comments concerning Major changes to All Reviewers**

Dear Reviewers,

Thanks a lot for reviewing our paper and for your valuable comments.

Before detailing the changes we have made to address your comments, we would like to highlight major changes we have brought to our work:

- Instead of re-training the GNN model, we propose a **fine-tuning strategy**. This enables us to overcome the OOD problem but also to make GInX-Eval, our evaluation method, applicable to black-box models accessible only via API calls, e.g. many LLMs.
- We have added a concrete example to illustrate the OOD problem (Figure 2).
- We have added the pseudocode of GInX-Eval in the Appendix.
- We have added Figure 1 to illustrate the general procedure and better illustrate our paper.
- We changed the figures in the Experimental section to display directly the GInX score instead of the Test Accuracy.
- We have uniformed the notations and used them consistently throughout the paper which improves the clarity of the Method section. For instance, we introduce h the explainability method, and chi the edge selection function…

To incorporate your feedback, we have uploaded a revision of our paper with the following main changes:

- We have re-defined the EdgeRank score to a new score the **HomophilicRank score**
- We have added more GNN models: **GCN**, GAT, and GIN models have been tested and the results are in the Appendix.
- With our **fine-tuning strategy**, we address all reviewers’ concerns about the usability of GInX-Eval for black-box models.

Significant changes to improve the clarity and methodological rigor have been made in the newly submitted version of the paper.

The main modifications are **highlighted in yellow in the newly submitted version**.

For minor or more specific comments, we invite each reviewer to read our targeted answers (see below) where we address their concerns in more detail.

---

### Author Response · Authors · 2023-11-21
**Final Discussion Time**

Dear Reviewers and Area Chair,

The rebuttal phase ends tomorrow and I haven't received any feedback from two reviewers. I am writing to you to engage in a constructive discussion before the deadline. I hope you had the time to go through **the many changes** I have made in the **main paper** and **supplementary material**, taking into account all your comments + adding many other things to enhance the clarity and quality of the paper.

Looking forward to hearing back from you,

Best Regards, the authors

---

### Meta-Review · Area_Chair_DiyZ · 2023-12-15

**Metareview:**

This paper proposes a new evaluation procedure for explainability methods of GNNs. It tackles an important and challenging question, i.e., proper evaluation of GNN explanations, and provides helpful insights on the limitations of existing metrics. However, as the reviewers pointed out, the proposed method is a straightforward adaptation of Hooker et al. (2018). The unique challenge of applying this method on GNNs isn't clear, and the authors didn't convincingly address this question in the response.

**Justification For Why Not Higher Score:**

Limited technical novelty.

**Justification For Why Not Lower Score:**

N/A

---

### Decision · Program_Chairs · 2024-01-16

Reject